

# Mechanistic pathways of mercury removal from the organomercurial lyase active site

Pedro J. Silva and Viviana Rodrigues

FP-ENAS/Fac. de Ciências da Saúde, Universidade Fernando Pessoa, Porto, Portugal

## ABSTRACT

Bacterial populations present in Hg-rich environments have evolved biological mechanisms to detoxify methylmercury and other organometallic mercury compounds. The most common resistance mechanism relies on the $H^+$-assisted cleavage of the Hg–C bond of methylmercury by the organomercurial lyase MerB. Although the initial reaction steps which lead to the loss of methane from methylmercury have already been studied experimentally and computationally, the reaction steps leading to the removal of $Hg^{2+}$ from MerB and regeneration of the active site for a new round of catalysis have not yet been elucidated. In this paper, we have studied the final steps of the reaction catalyzed by MerB through quantum chemical computations at the combined MP2/CBS//B3PW91/6-31G(d) level of theory. While conceptually simple, these reaction steps occur in a complex potential energy surface where several distinct pathways are accessible and may operate concurrently. The only pathway which clearly emerges as forbidden in our analysis is the one arising from the sequential addition of two thiolates to the metal atom, due to the accumulation of negative charges in the active site. The addition of two thiols, in contrast, leads to two feasible mechanistic possibilities. The most straightforward pathway proceeds through proton transfer from the attacking thiol to Cys159 , leading to its removal from the mercury coordination sphere, followed by a slower attack of a second thiol, which removes Cys96. The other pathway involves Asp99 in an accessory role similar to the one observed earlier for the initial stages of the reaction and affords a lower activation enthalpy, around 14 kcal mol$^{-1}$, determined solely by the cysteine removal step rather than by the thiol ligation step. Addition of one thiolate to the intermediates arising from either thiol attack occurs without a barrier and produces an intermediate bound to one active site cysteine and from which $Hg(SCH_3)_2$ may be removed only after protonation by solvent-provided $H_3O^+$. Thiolate addition to the active site (prior to any attack by thiols) leads to pathways where the removal of the first cysteine becomes the rate-determining step, irrespective of whether Cys159 or Cys96 leaves first. Comparisons with the recently computed mechanism of the related enzyme MerA further underline the important role of Asp99 in the energetics of the MerB reaction. Kinetic simulation of the mechanism derived from our computations strongly suggests that *in vivo* the thiolate-only pathway is operative, and the Asp-assisted pathway (as well as the conversion of intermediates of the thiolate pathway into intermediates of the Cys-assisted pathway) is prevented by steric factors absent from our model and related to the precise geometry of the organomercurial binding-pocket.

Corresponding author
Pedro J. Silva, pedros@ufp.edu.pt

## INTRODUCTION

Mercury is naturally present in the environment, especially at specific geologically enriched regions along tectonical plate boundaries (*Varekamp & Buseck, 1986*), where it can be found as the characteristically colored cinnabar ores (HgS). Though quite insoluble in water ($\approx$10 µg/L), the solubilized species ($Hg^{2+}$) may be readily uptaken by methanogens and sulfate-reducing bacteria, which then methylate it to methylmercury (*Barkay, Miller & Summers, 2003*; *Lin, Yee & Barkay, 2012*) through the combined action of the reductive acetyl-CoA pathway (*Choi, Chase & Bartha, 1994*) and two novel proteins: a methyl-binding corrinoid-containing protein (HgcA) and a corrinoid-reducing protein with unknown physiological function(*Parks et al., 2013*). The methylmercury thus formed is highly soluble in lipids and therefore tends to accumulate in living tissues and to be concentrated along the food chain. The solution reactivity of mercury towards soft ligands (*Riccardi et al., 2013*) like the thiols present in cysteine-containing proteins is responsible for the high toxicity of methylmercury (*Eto, Marumoto & Takeya, 2010*), especially towards lipid-enriched cells (like those of the nervous system) where its solubility is the highest. Bacterial populations present in Hg-rich environments have therefore evolved biological mechanisms to detoxify methylmercury and other organometallic mercury compounds. The most common resistance mechanism relies on proton-assisted cleavage of the Hg–C bond of methylmercury by the organomercurial lyase MerB (*Begley, Walts & Walsh, 1986a*; *Begley, Walts & Walsh, 1986b*), and sequential transfer of the remaining $Hg^{2+}$ ion to a flavoprotein (MerA) which reduces the cation to its metallic form (*Fox & Walsh, 1982*; *Ledwidge et al., 2005*; *Ledwidge et al., 2010*).

Extensive experimental studies (*Begley, Walts & Walsh, 1986a*; *Begley, Walts & Walsh, 1986b*; *Pitts & Summers, 2002*; *Di Lello et al., 2004*; *Lafrance-Vanasse et al., 2009*) have elucidated the structure of MerB and established that this enzyme does not require any cofactors and uses two thiols (like cysteine or glutathione, but not dithiothreytol (*Pitts & Summers, 2002*)) as co-reactants for every mercury organic compound cleaved. A pioneering computational study (*Parks et al., 2009*) has shown that in the active site any one of two conserved Cys residues (Cys 96 and Cys 159) may, upon deprotonation, complex the Hg moiety of the substrate. A proton is then transferred from the other conserved Cys to a conserved acidic residue (Asp 99), which subsequently acts as a proton donor to the leaving alkyl or aryl group. That study did not, however, address the reaction steps leading to the loss of $Hg^{2+}$ from MerB and regeneration of the active site for a new round of catalysis. In this contribution, we use quantum chemical computations to analyze the possible reaction mechanisms. The application of these methods to protein active sites (reviewed e.g., in *Himo & Siegbahn, 2003*; *Ramos & Fernandes, 2008*) allows the characterization of enzyme-bound intermediates and transition states which (due to their transient nature) are not amenable to experimental characterization and the evaluation

of thermodynamic and kinetic feasibility of postulated mechanisms. The results show that complexation of $Hg^{2+}$ by extraneous thiols is most likely to proceed through the attack by one protonated and one deprotonated thiol, rather than by two deprotonated thiols. The precise order of attack seems to be arbitrary, and an important role of the $Hg^{2+}$-coordinating Asp99 residue in lowering the reaction energy was found.

## METHODS

The active site geometry was built from PDB:3F0P, the crystal structure of the mercury-bound form of MerB (*Lafrance-Vanasse et al., 2009*). The active site included the conserved residues Cys96, Asp 99, Cys 159, the mercury ion and Hg-complexing water molecule. To prevent unrealistic motions of the active site, the $C\alpha$ and $C\beta$ atoms of every aminoacid were constrained to their crystallographic positions. All calculations were performed at the B3PW91 level of theory (*Perdew, 1991*; *Becke, 1993*), which has been commonly used in the study of Hg-containing complexes (*Barone et al., 1997*; *Ni et al., 2006*; *Parks et al., 2009*; *Li et al., 2010*; *Riccardi et al., 2013*). Autogenerated delocalized coordinates (*Baker, Kessi & Delley, 1996*) were used for geometry optimizations, using the SDD effective core-potential and associated basis set (*Häussermann et al., 1993*) for Hg and the 6-31G(d) basis set for all other atoms. More accurate DFT energies of the optimized geometries were calculated with a triple-$\zeta$ quality basis set, 6-311 + G(d). Zero point (ZPE) and thermal effects ($T = 298.15$ K, $P = 1$ bar) were evaluated using a scaling factor of 0.9804 for the computed frequencies. All computations were performed with the Firefly quantum chemistry package, which is partially based on the GAMESS (US) (*Schmidt et al., 1993*) source code. Environmental contributions to the energies of the stationary points and transition states were computed with the polarizable conductor model (*Tomasi & Persico, 1994*; *Mennucci & Tomasi, 1997*; *Cossi et al., 1998*), with dielectric constants ranging from 4 (usually chosen for protein-embedded active sites) to 78.36 (mimicking a completely exposed active site). Dispersion and repulsion effects were evaluated as described by Amovilli and Mennucci (*Amovilli & Mennucci, 1997*). MP2 single-point energies were computed on the optimized geometries using the aug-cc-pVDZ-PP (or aug-ccpVTZ-PP) basis set (*Peterson & Puzzarini, 2005*) for mercury and cc-pVDZ (or cc-pVTZ basis sets) for all other elements, and extrapolated to the complete basis set limit (CBS-MP2) as described by Truhlar (*Truhlar, 1998*). Solution MP2 values were obtained by applying the DFT solvation energies to the gas-phase CBS-MP2 energies. Numerical integration of the rate equations of the kinetic model derived in this work was performed using a self-built program, which is available for download, together with respective outputs at http://dx.doi.org/10.6084/m9.figshare.1433993.

## RESULTS

A large number of mechanistic pathways for $Hg^{2+}$ removal from the active site of MerB is possible (Fig. 1), depending on the protonation state of each mercury-attacking ligand (thiol vs. thiolate), on whether Cys96 or Cys159 is first ejected from the coordination sphere of the Hg ion, and on whether the protonation state of Asp99 changes throughout the cycle. Our density-functional computations show that extraneous methanethiol is

**Figure 1 Pathways for Hg removal from MerB, starting from an attacking thiol ("thiol-based" mechanism) or an attacking thiolate ("thiolate-based" mechanism).** In both mechanisms, primed-numbered intermediates arise from the attack of a thiol and a thiolate, whereas intermediates numbered with unprimed numbers arise from the attack of two species with the same protonation state (either two thiols or two thiolates).

not nucleophilic enough to directly the attack of the enzyme-bound $Hg^{2+}$. The moderate acidity of the thiol, however, allows it to transfer a proton to one of the $Hg^{2+}$ ligands (either Cys159 or Asp 99), in a process which both weakens the ligand-to-metal bond and transforms the thiol into a (much more nucleophilic) thiolate (Fig. 2). Proton transfer to Cys159 (Fig. 2B) occurs with a small barrier (12.3–12.8 kcal mol$^{-1}$ in MP2, 7.8–8.0 kcal mol$^{-1}$ using DFT) and is moderately exergonic by 7–9 kcal mol$^{-1}$. This activation barrier is very similar to the barrier found experimentally (*Hong et al., 2010*) for the initial attack of MerB-bound mercury by free glutathione ($2.5 \times 10^4$ M$^{-1}$ s$^{-1}$, which translates to 11.4 kcal mol$^{-1}$).

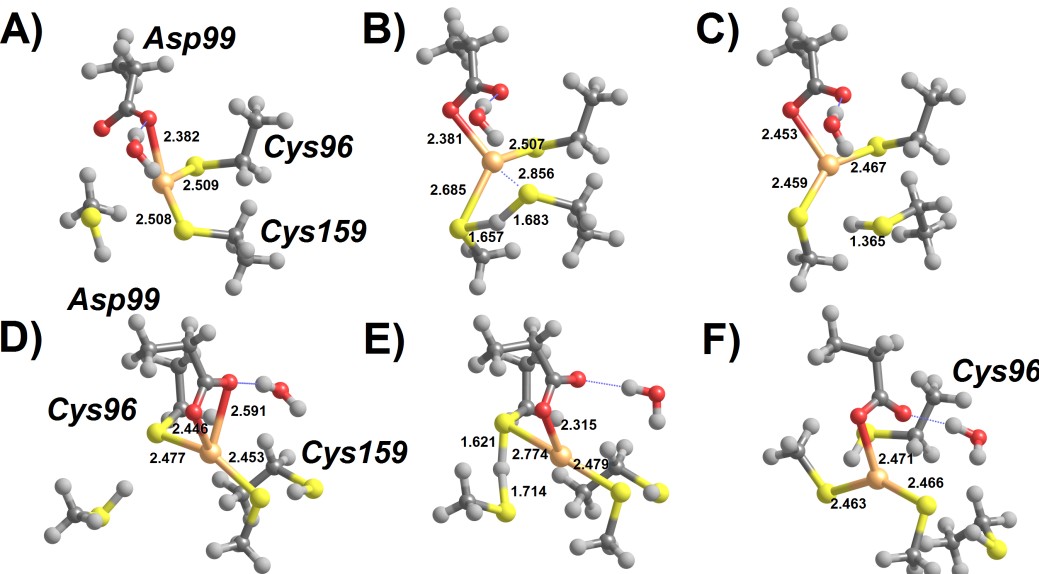

**Figure 2 Cys-assisted thiol addition to Hg²⁺.** (A) Pre-reactional complex (Int1); (B) H⁺ transfer to Cys 159 (transition state); (C) thiol-based Int2 (Cys96-bound); (D) thiol-based Int2 (Cys96-bound) + CH₃SH; (E) H⁺ transfer to Cys 96 (transition state); (F) thiol-based Int4 (Asp99-bound). Relevant distances (in Ångstrom) are highlighted. Molecules (D–E) have been rotated counterclockwise ca. 90° around the z-axis, relative to the orientation of molecules (A–C).

The addition of a second thiol to the singly-cysteinated Hg²⁺ is quite similar to that of the first thiol, as expected from the identical composition of coordination sphere around the metal atom (a carboxylate and two thiols). The most interesting difference arises from the possibility of proton transfer to Cys96 (in the Cys96-bound Int2) due to the newly-found flexibility of the freed Cys159 sidechain. This step (Figs. 2D–2F) has a larger barrier (15.8–18.1 kcal mol⁻¹ using MP2, 14.5–14.8 kcal mol⁻¹ in DFT) than the addition of the first thiol because the larger thiol(ate)-Hg distance in the latter transition state (2.875 vs. 2.685 Å) entails a smaller stabilization due to lower overlap between thiol(ate) and Hg orbitals. In the gas phase, regeneration of the active site through the removal of Hg(SCH₃)₂ from Asp99 leads to a continuous increase in electronic energy of approximately 26 kcal mol⁻¹. In solution, however, the reaction is only moderately endergonic (1–6 kcal mol⁻¹, depending on the dielectric constant) since the presence of a compact negative charge in the Asp99 residue in the product state leads to a stronger solvation of the separated fragments, which largely offsets the gas-phase energy increase due to the severing of the Hg-carboxylate bond.

If the initial conformation of the attacking thiol, in contrast to that depicted in Fig. 2, has the S–H bond aligned towards Asp99, H⁺-transfer to Asp99 occurs instead, without any thermodynamic barrier (Fig. 3A). This transfer is favorable by 15 kcal mol⁻¹ and may be followed by a further movement of the proton from Asp99 to the distal Cys96 Hg-ligand (Fig. 3B), which is thus released from the metal (Fig. 3C). This proton-transfer step has a moderate barrier around 12–14 kcal mol⁻¹, and should therefore occur at a rate similar to that of the direct protonation and removal of the Cys159 ligand depicted in the alternative
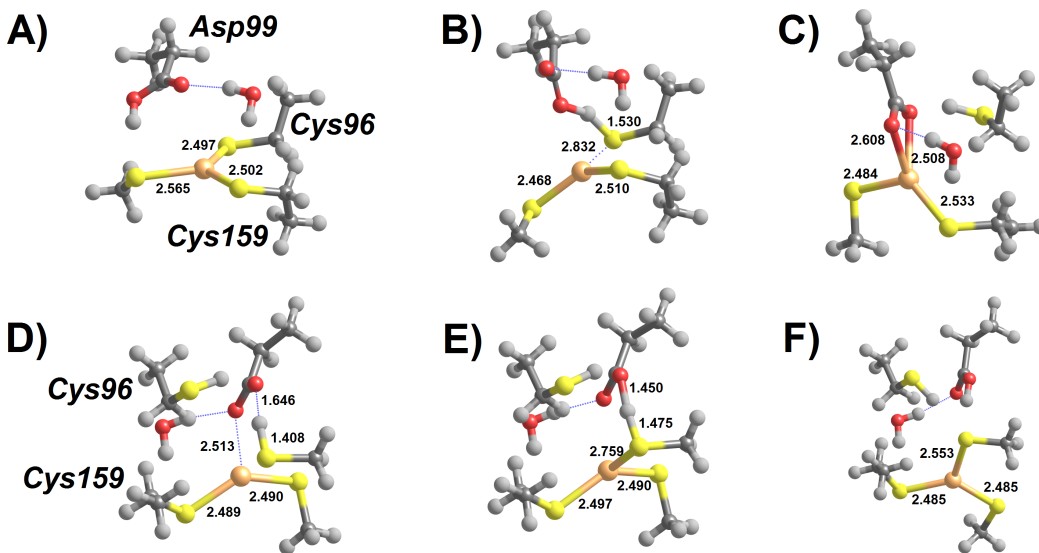

**Figure 3 Asp99-assisted thiol addition to Hg²⁺.** (A) Asp 99 receives H⁺ from the attacking thiol;
(B) H⁺ transfer from Asp99 to Cys96 (transition state); (C) thiol-based Int2 (Cys159-bound); (D)
thiol-based Int2 (Cys159-bound) + CH₃SH; (E) H⁺ transfer from thiol to Asp99 (transition state);
(F) thiol-based Int4 (Cys159-bound). Molecules (D–F) are depicted as seen from a point of view
approximately opposite that used in the depiction of molecules (A–C). Relevant distances (in Ångstrom)
are highlighted.

mechanism above (Figs. 2A–2C). The addition of a second thiol may again proceed in an
Asp99-assisted fashion (Figs. 3D–3F): proton transfer from the thiol to the Asp99 ligand
of the Cys159-bound Int2 is favored by 10–11 kcal-mol⁻¹ but must now overcome a
small barrier (4 kcal mol⁻¹), in contrast to the barrier-free process observed when this
movement is the first step of the reaction sequence.

In contrast to the addition of cysteine thiols analyzed above, addition of a cysteine
thiolate to the MerB-bound Hg²⁺ (Fig. 4) proceeds unhindered, i.e., without any energetic
barrier. The tetra-coordinated intermediate formed (Int1) lies 12–13 kcal mol⁻¹ below
the infinitely-separated reactants (in MP2; 6–7 kcal mol⁻¹ below reactants in DFT),
and may then shed any of its Cys-ligands upon overcoming a moderate 14.0–15.5 kcal
mol⁻¹ barrier. The addition of a second thiolate to this complex, however, is much
costlier due to the electrostatic repulsion between the freed, deprotonated, Cys and the
negatively-charged thiolate. The precise cost depends very steeply on the chosen dielectric
constant (Table 3), as expected for a reaction involving highly localized charges, but the
transition state for this step always remains more than 25 kcal mol⁻¹ above Int1, far above
the 16–20 kcal mol⁻¹ expected (*Parks et al., 2009*) for the rate-limiting step of this enzyme
from the application of the Eyring equation, $k_{cat} = \frac{k_B T}{h} e^{-\frac{\Delta G^{\ddagger}}{RT}}$, to the experimentally
observed reaction rate (*Begley, Walts & Walsh, 1986b*). Since methanethiol is weakly acidic
(pKa = 10.4) and is almost completely protonated at physiological pH, the initial cost of
deprotonating it must be added to the computations above. This cost amounts to $-RT$ ln
$10^{pH-pKa}$ (*Ullmann & Knapp, 1999*), or 4.6 kcal mol⁻¹ at pH = 7.

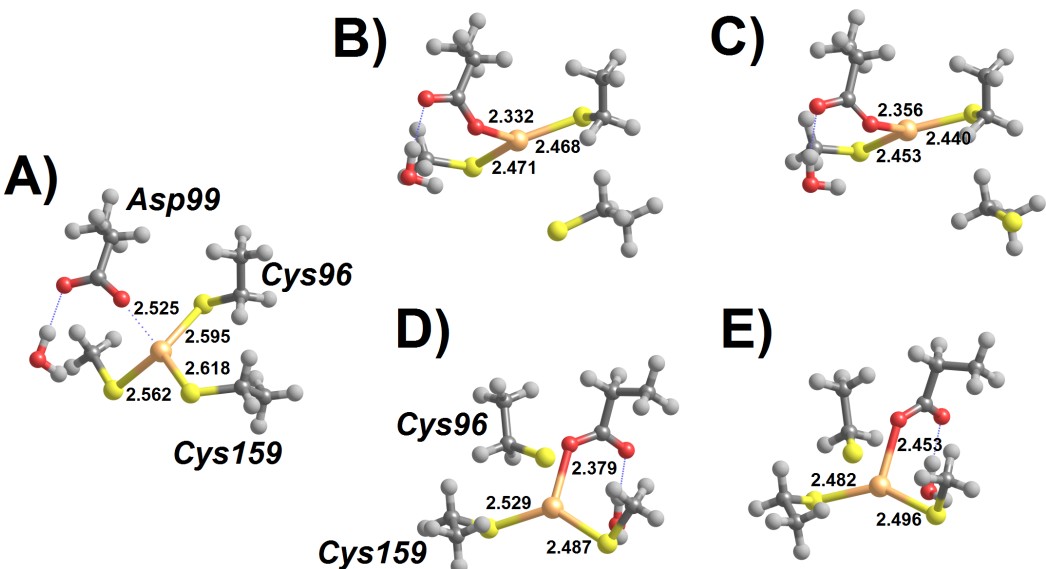

**Figure 4 Addition of deprotonated thiol to MerB-bound Hg²⁺.** (A) Thiolate-based Int1; (B) breaking the Cys159-Hg bond (transition state); (C) thiolate-based Int2 (C96-bound); (D) breaking the Cys96-Hg bond (transition state); (E) thiolate-based Int2 (C159-bound). Molecules (D–E) are depicted as seen from a point of view approximately opposite that used in the depiction of molecules (A–C). Relevant distances (in Ångstrom) are highlighted.

So far, we have only described the reaction mechanism arising from the addition of two thiols with the same protonation state. We now turn to the analysis of mechanism involving distinct protonation states of the attacking thiols: inded, the two N-terminal cysteines of MerA which catalyze removal of $Hg^{2+}$ from MerB *in vivo* (*Ledwidge et al., 2005*) have been shown to possess widely separated $pK_a$'s (*Ledwidge et al., 2010*) which entail that at physiological pH one of them is expected to remain mostly unprotonated while the other only deprotonates at high pH.

The addition of a thiolate to any of the forms of thiol-based intermediate 2 (where $Hg^{2+}$ is bound to either of Cys159 or Cys96) occurs spontaneously without any energetic barrier. In the Cys159-bound form, the reaction product has a slightly lower energy than in the Cys96-bound form and adopts a more exposed conformation (Fig. 5D). The metal ion in the resulting intermediate 3′ has a sulfur-only coordination sphere in both instances, as the interactions with Asp99 have disappeared (Figs. 5B and 5D).

Reaction of a thiol with the *thiolate*-based C96-bound/C159-deprotonated intermediate proceeds readily through proton transfer from the attacking thiol to the deprotonated Cys159 and immediate thiolate attack of the metal atom. This process occurs without an electronic barrier and yields C96-bound Int3′ (Fig. 5B). On the other hand, the *thiolate*-based C159-bound intermediate 2 is susceptible to attack by a thiol in an Asp99-dependent manner (Fig. 5F). As in the other Asp99-assisted thiol attacks analyzed above, the electronic barrier to this process is negligible (Fig. 5E) and yields an intermediate where Asp99 is protonated and the mercury ion remains coordinated by three ligands (two external thiolates and Cys159). This Cys159-bound/Cys96-deprotonated/Asp99-protonated

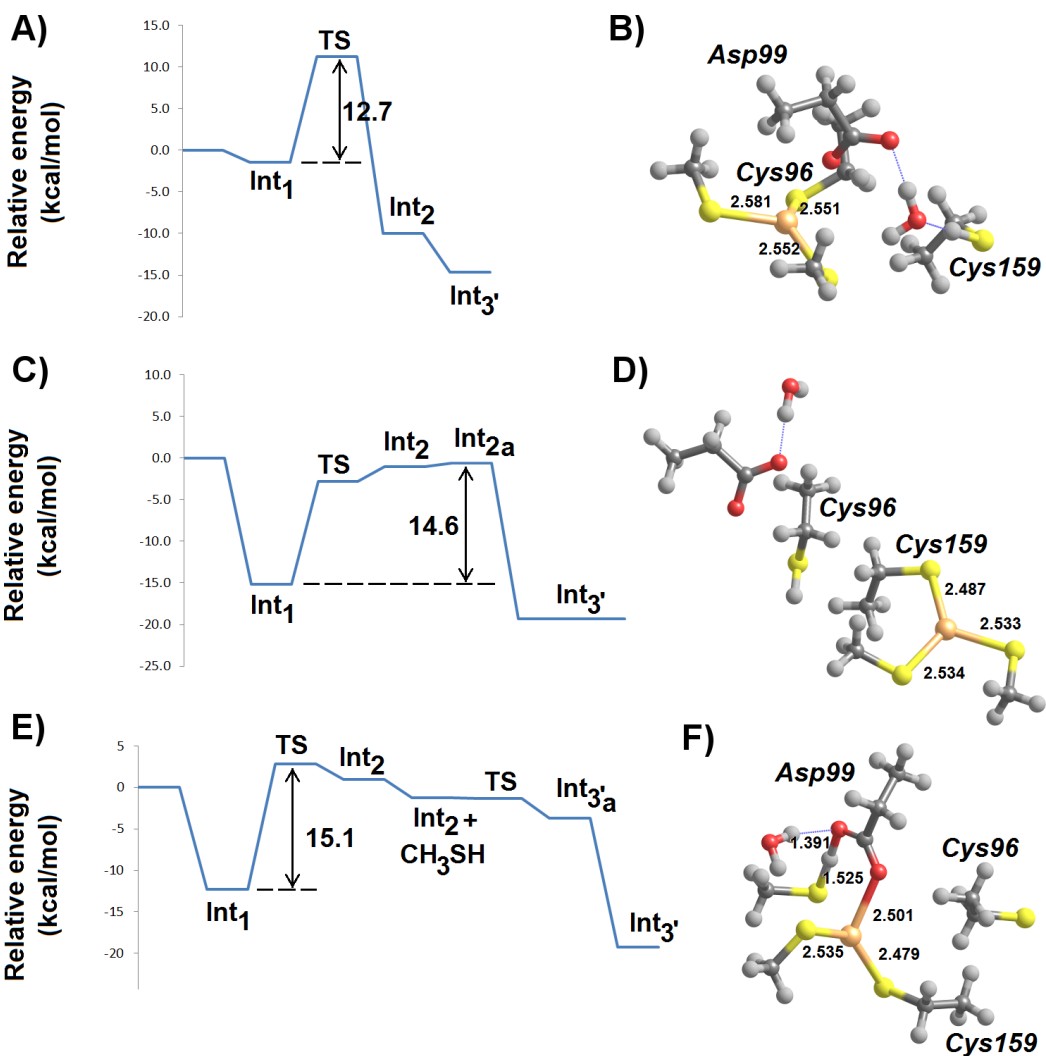

**Figure 5** MP2/CBS//B3PW91/6-31G(d) energetic profiles (with $\varepsilon = 20$) and representative structures of intermediates arising from attack of $Hg^{2+}$ by a thiol and a thiolate. (A) Energetic profile of Cys159-assisted thiol attack followed by thiolate addition; (B) structure of $Int3'$ arising from Cys159-assisted thiol attack; (C) energetic profile of Asp99-assisted thiol attack followed by thiolate addition; (D) structure of $Int3'$ arising from Asp99-assisted thiol attack followed by thiolate addition; (E) energetic profile of an initial thiolate attack followed by Asp99-assisted thiol addition to $Hg^{2+}$; (F) structure of the transition state of Asp99-assisted thiol addition to thiolate-based Int 2. Relevant distances (in Ångstrom) are highlighted.

intermediate ($Int3'_a$ in Fig. 5E) spontaneously decays, through a negligible energetic barrier ($<1$ kcal mol$^{-1}$), to the Cys159 bound/ Cys96-protonated/Asp99-deprotonated state.

The overall reaction barrier for all of the mechanisms involving attack of the $Hg^{2+}$ ion by a thiol and a thiolate therefore depends on the barrier of the first attack, which was computed above (Tables 1–3) to lie between 12 and 15 kcal mol$^{-1}$ in all instances. Since the 3–4 kcal mol$^{-1}$ difference between the barriers of these alternatives is equivalent to the intrinsic error of the computational protocols used, further discrimination between these three possibilities is unfortunately not possible at this stage.

**Table 1** Relative enthalpies (kcal mol$^{-1}$) of the reaction intermediates in the Cys-assisted thiol addition to MerB-bound Hg$^{2+}$, computed at the MP2/CBS//B3PW91/6-31G(d) level of theory.

| | $\varepsilon = 4$ | $\varepsilon = 10$ | $\varepsilon = 20$ | $\varepsilon = 78.36$ |
|---|---|---|---|---|
| Reagent + CH$_3$SH | 0.0 | 0.0 | 0.0 | 0.0 |
| Int 1 | −0.6 | −1.2 | −1.5 | −1.7 |
| TS Int1 to Int2 (Cys-96 bound) | 11.7 | 11.4 | 11.3 | 11.1 |
| Int2 (Cys-96 bound) | −8.3 | −9.3 | −10.0 | −10.8 |
| Int2 (Cys-96 bound) + CH$_3$SH | −11.8 | −12.4 | −13.1 | −14.0 |
| TS Int2 to Int4 | 4.1 | 4.4 | 4.4 | 4.2 |
| Thiol-based Int4 (Asp99-bound) | −12.8 | −12.8 | −13.5 | −14.3 |
| Infinitely separated products | −7.3 | −10.7 | −12.1 | −13.3 |

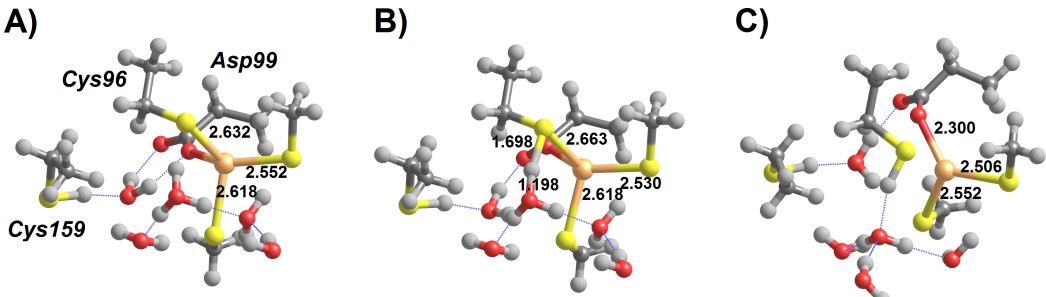

**Figure 6** H$_3$O$^+$-assisted removal of Hg(SCH$_3$)$_2$ from the MerB active site (compact conformation). (A) Cys96-bound Int3′ surrounded by water-solvated H$_3$O$^+$; (B) proton transfer from H$_3$O$^+$ to Cys96 (transition state); (C) Asp-bound Hg(SCH$_3$)$_2$ (Int4). Relevant distances (in Ångstrom) are highlighted.

Regeneration of the initial state of the active site from the Int3′ intermediate now requires the severing of the remaining Hg–Cys bond. Preliminary attempts at the characterization of this reaction step showed that direct stretching of the Hg–Cys bond is energetically quite costly. Our results above (Table 2), however, show that protonation of the metal-bound Cys dramatically weakens the Hg–S bond. We have therefore analyzed the feasibility of removing Hg(SCH$_3$)$_2$ from the active site cysteine through direct protonation by solvent-provided H$_3$O$^+$. A few explicit water molecules were also added to the model to provide an appropriate description of the solvated hydronium ion (Fig. 6).

As mentioned above, two different conformations of the Int3′ intermediate exist: an extended conformation (Fig. 5D) where Hg(SCH$_3$)$_2$ is bound to Cys159 and a compact conformation where the product is bound to Cys96, instead (Fig. 5B). In the compact conformation (Fig. 6) this proton transfer is spontaneous by 6.6 kcal mol$^{-1}$ (according to MP2; 2.7 kcal mol$^{-1}$ according to DFT) and diffusion-controlled: the very small energetic barrier found during the geometry optimization completely disappears upon inclusion of solvation, zero-point and vibrational effects. Upon removal of Cys96, Asp99 weakly attaches to the mercury ion, preventing the product from freely diffusing away from the

**Table 2** Relative enthalpies (kcal mol$^{-1}$) of the reaction intermediates in the Asp99-assisted thiol addition to MerB-bound Hg$^{2+}$, computed at the MP2/CBS//B3PW91/6-31G(d) level of theory.

| | $\varepsilon = 4$ | $\varepsilon = 10$ | $\varepsilon = 20$ | $\varepsilon = 78.36$ |
|---|---|---|---|---|
| Reagent +CH$_3$SH | 0.0 | 0.0 | 0.0 | 0.0 |
| Int 1 (protonated Asp99) | −15.4 | −15.2 | −15.2 | −15.2 |
| TS Int1 → Int2 (H$^+$ moves from Asp99 to Cys96) | −1.7 | −2.4 | −2.8 | −3.2 |
| Int 2 (Cys-159 bound) | −1.0 | −0.9 | −1.0 | −1.1 |
| Int 2 (Cys-159 bound) + CH$_3$SH | −9.6 | −8.6 | −8.3 | −8.2 |
| TS Int2 (Cys-159 bound) to Int4 (Cys159-bound) | −5.5 | −4.7 | −4.5 | −4.4 |
| Thiol-based Int4 (Cys159-bound) | −19.8 | −19.4 | −19.4 | −19.5 |

**Table 3** Relative enthalpies (kcal mol$^{-1}$) of the reaction intermediates in thiolate addition to MerB-bound Hg$^{2+}$, computed at the MP2/CBS//B3PW91/6-31G(d) level of theory.

| | $\varepsilon = 4$ | $\varepsilon = 10$ | $\varepsilon = 20$ | $\varepsilon = 78.36$ |
|---|---|---|---|---|
| Reactant + CH$_3$S$^-$ | 0.0 | 0.0 | 0.0 | 0.0 |
| Thiolate-based Int1 | −13.1 | −12.6 | −12.3 | −12.1 |
| Thiolate-based TS 1 →2 (C96-bound) | 1.8 | 1.8 | 1.8 | 1.9 |
| Thiolate-based Int2 (C96-bound) | −3.7 | −2.6 | −2.2 | −1.9 |
| Thiolate-based Int2 (C96-bound) + CH$_3$S$^-$ TS | 26.8 | 17.1 | 14.0 | 11.8 |
| Thiolate-based Int3 (C96-bound) | 10.6 | 1.5 | −1.3 | −3.2 |
| Reactant + CH$_3$S$^-$ | 0.0 | 0.0 | 0.0 | 0.0 |
| Thiolate-based Int1 | −13.1 | −12.6 | −12.3 | −12.1 |
| Thiolate-based TS 1 →2 (C159-bound) | 1.5 | 2.4 | 2.8 | 3.2 |
| Thiolate-based Int2 (C159-bound) | −0.6 | 0.5 | 1.0 | 1.5 |
| Thiolate-based Int2 (C159-bound) + CH$_3$S$^-$ TS | 32.3 | 21.0 | 17.3 | 14.6 |
| Thiolate-based Int3 (C159-bound) | 14.3 | 3.4 | −0.1 | −2.5 |

active site. Complete removal of Hg(SCH$_3$)$_2$ occurs upon stretching this very weak Asp-Hg bond, at a cost of only 4.1 kcal mol$^{-1}$.

In the "extended" conformation of Int3′, the Hg(SCH$_3$)$_2$ moiety lies quite far from Asp99, which modifies the mechanistic analysis due to the impossibility of Asp99-attachment to the metal upon the release of Cys159. In contrast to the previous analysis, in this conformation the solvated H$_3$O$^+$ is unstable even before including bulk solvation effects implicitly through the PCM model. Instead, two separate minima arise: an unproductive intermediate featuring a proton on the Asp99 residue (Fig. 7A), and the Cys159-protonated product featuring a free Hg(SCH$_3$)$_2$ (Fig. 7C). Both minima lie ≈10 kcal mol$^{-1}$ below the postulated initial (meta-stable) conformation featuring a solvated H$_3$O$^+$ (Fig. 7B).

## Kinetic simulations

Extensive experimental analysis of the reaction of Hg-bound MerB with glutathione or the physiological partner (*Hong et al., 2010*) has shown that MerA is able to effect

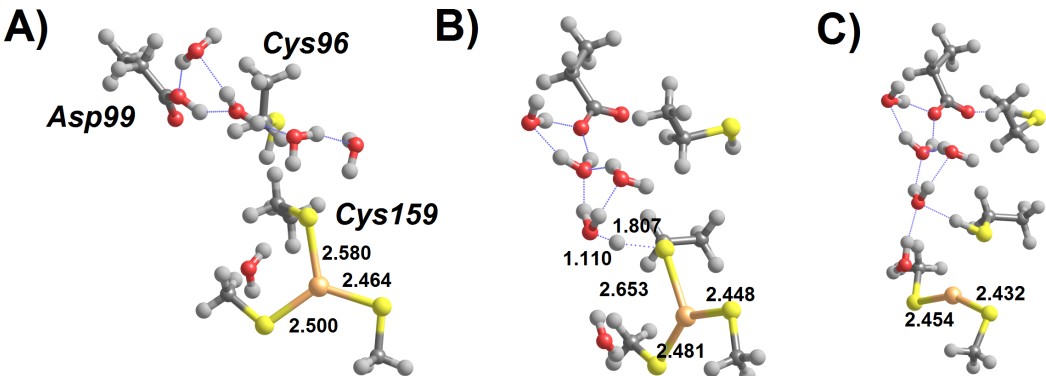

**Figure 7 H₃O⁺-assisted removal of Hg(SCH₃)₂ from the MerB active site (extended conformation)** (A) Asp99-protonated Int3′ surrounded by water molecules; (B) proton transfer from Asp99 to Cys159 (transition state); (C) regenerated active site with released Hg(SCH₃)₂. Relevant distances (in Ångstrom) are highlighted.

complete metal removal even at very low concentrations (50 μM), whereas concentrations of monothiols below 10 mM afford only partial protein demetallation. Numerical simulation of the complete reaction mechanism described in this work (Fig. 8) reveals a very good agreement with experiment, provided that a protonated thiol is prevented from performing the initial attack on the mercury ion (Fig. 9A and 9B): operation of the Asp-assisted pathway (either alone or in concert with other pathways) would always lead to complete removal of mercury from the MerB active site (Fig. 9C) due to the high exergonicity of the initial formation of the Asp-protonated form of Int1 intermediate (Fig. 3A and Table 2). Simultaneous operation of the Cys-assisted pathways would in turn allow the C96-bound Int3′ intermediate (formed mainly in the thiolate pathway, which has a more exergonic first reaction than the Cys-assisted thiol attack pathway) to be diverted through thiolate loss (reaction $k_6$ in Fig. 8) to the Cys-assisted pathway, yielding a complex kinetic profile which ultimately leads to total mercury removal from MerB (Fig. 9D). In turn, setting the reaction rate of the $k_5$ and $k_6$ steps to zero (i.e., preventing the conversion of Int2 (C96-bound) into Int3′ (C96-bound, and vice-versa)), while keeping the thiolate-only pathway and the rest of the Cys-assisted pathway operative yields a kinetic profile indistinguishable from that of the thiolate-only pathway. Interestingly, identical kinetic simulations using the DTT (which is a known inhibitor of MerB) failed to show any inhibition. The agreement of our model with the experimental observations therefore requires that the formation of Int1 (protonated Asp) (Fig. 8, reaction $k_{15}/k_{16}$), the conversion of Int2 (C96-bound) into Int3′ (C96-bound) (Fig. 8, reaction $k_5/k_6$), and the release of the Hg-DTT complex from the active site, which are predicted by our small-model QM computations to be thermodynamically and kinetically feasible, are prevented in the enzyme, most likely due to the intervention of steric factors arising from the rest of the protein. The proposed role of steric factors in the overall kinetic profile of MerB is consistent with other experimental observation: for example, though the trigonal complex of Hg bound by both sulfur atoms of DTT and by Cys96 is long-lived in the absence of added thiols, Hg can be removed after a few minutes of incubation with MerA or

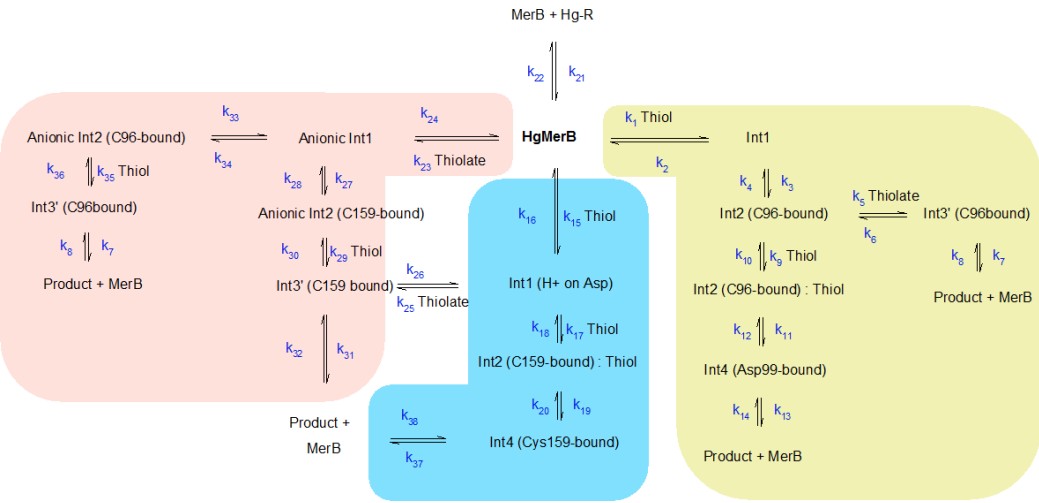

**Figure 8 Complete reaction mechanism of the mercury-removing steps in the MerB reaction cycle.** Thiolate-based pathways are depicted in salmon; Asp-assisted thiol addition in sapphire blue; Cys-assisted thiol addition in light olive. Communication between the thiolate-based and the Cys-assisted pathways is not shown explicitly (in contrast to that between thiolate-based and Asp-assisted pathways) but is possible due to the presence of the common intermediate, Int3' (C96bound).

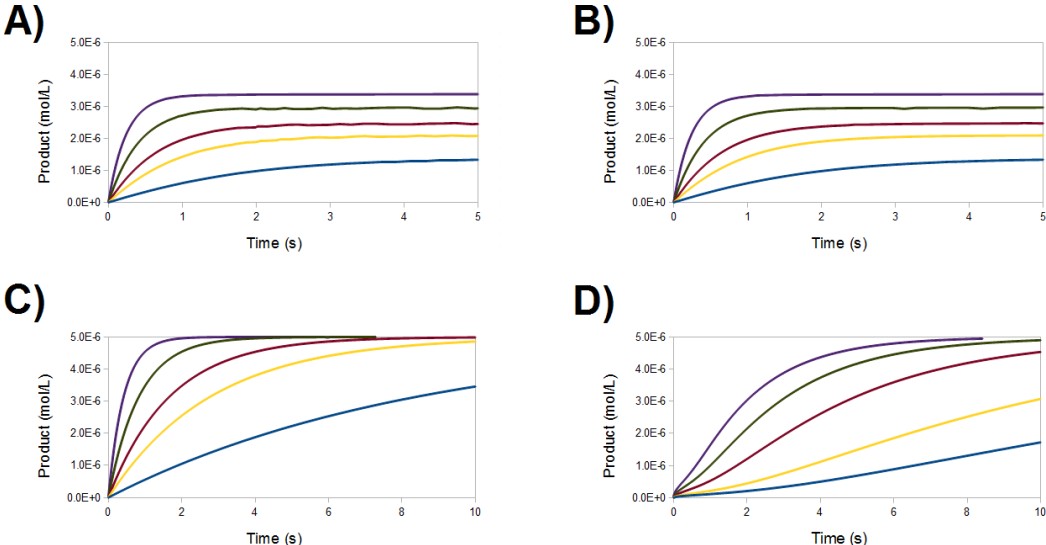

**Figure 9 Numerical simulations of different portions of the MerB reaction mechanism in the presence of glutathione, using reaction rates derived from the energies computed by our quantum chemical computations.** (A) Thiolate-based pathways only; (B) thiolate-based pathways + conversions to intermediates of the Asp-assisted pathway. Rates of reactions $k_{15}/k_{16}$ set to zero; (C) combined operation of thiolate-based pathways and Asp-assisted thiol addition pathways; (D) thiolate-based pathways + Cys-assisted thiol addition + conversions to intermediates of the Asp-assisted pathway. Rates of reactions $k_{15}/k_{16}$ set to zero. Glutathione concentrations are: 0.5 mM (blue), 1.5 mM (yellow), 2.5 mM (red), 5 mM (green), and 10 mM (purple)

with incubation with very high concentrations of glutathione (*Benison et al., 2004*). Since the chemically reactive portion in all these MerB co-substrates is the same, this implies that the differences in behavior should not be based in purely electronic factors but in the presence of intermolecular interactions between Hg-DTT and MerB (but not between Hg-MerA and MerB, or between Hg-glutathione and MerB) which prevent its exit from the active site. Indeed, preliminary experimental evidence (J Omichinski, pers. comm., 2015) suggests that the N-terminal portion of MerB is responsible for trapping the Hg-DTT complex and the observed partial inhibition of MerB activity by DTT.

## DISCUSSION

Our computations show that the final steps of the reaction catalyzed by MerB, while conceptually simple, occur in a complex potential energy surface where several distinct pathways are accessible and may operate concurrently. The only pathway which clearly emerges as forbidden in our quantum chemical analysis is the one arising from the sequential addition of two thiolates to the metal atom, due to the accumulation of negative charges in the active site. The addition of two thiols, in contrast, leads to two feasible mechanistic possibilities. The most straightforward pathway proceeds through proton transfer from the attacking thiol to Cys159 (activation $\Delta H = 13 \, \text{kcal mol}^{-1}$), leading to its removal from the mercury coordination sphere, followed by a slower attack of a second thiol, which removes Cys96 (activation $\Delta H = 16\text{--}18 \, \text{kcal mol}^{-1}$). Entropic effects, which we could not analyze due to the need of enforcing geometric constraints on our active site model, may, however, easily place this pathway above the experimentally determined activation $\Delta G$ (16–20 kcal mol$^{-1}$). The other pathway involves Asp99 in an accessory role similar to the one observed earlier for the initial stages of the reaction (*Parks et al., 2009*) and affords a lower activation enthalpy, around 14 kcal mol$^{-1}$, determined solely by the removal of the first cysteine ligand rather than by the ligation of the second thiol. Unlike this Asp99-assisted mechanism, the Cys-assisted pathway predicts the addition of the second thiol to be rate-limiting, in marked disagreement with previous interpretations of experimental results (*Hong et al., 2010*). In contrast, addition of one thiolate to the intermediates arising from either thiol leads to pathways where the later reaction steps have negligible barriers. The intermediate formed in this reaction (Int3′) remains bound to one active site cysteine and may shed Hg(SCH$_3$)$_2$ after protonation of this cysteine by solvent-provided H$_3$O$^+$. The activation energy of this step in solution therefore depends on the solution pH according to equation:

$$\Delta G^{\ddagger} = \Delta G^{0\ddagger} + RT \ln \frac{1}{[\text{H}^+]}.$$

This protonation event is quite spontaneous and occurs without an energetic barrier ($\Delta G^{0\ddagger} = 0$), leading to an effective $\Delta G^{\ddagger} = -RT \ln 10^{-\text{pH}}$, or 9.5 kcal mol$^{-1}$ at pH = 7.

Thiolate addition to the active site (prior to any attack by thiols) leads to pathways where the removal of the first cysteine becomes the rate-determining step (activation $\Delta H = 14\text{--}15 \, \text{kcal mol}^{-1}$, irrespective of whether Cys159 or Cys96 leaves first). Asp99-assisted addition of a thiol to this intermediate then occurs without an energy barrier and yields the

familiar Int3′ intermediate discussed above. A comparison of these results with the recently published computational analysis of the transfer of $Hg^{2+}$ from the C-terminal cysteine pair of MerA to the buried cysteine pair in the active site of MerA (*Lian et al., 2014*) affords additional insights on the relative importance of the mercury-coordinating aminoacids. In that work, which included (unlike ours) the influence of the remainder of the enzyme through a QM/MM formalism, thiol addition to a $Hg^{2+}$ ion coordinated by two cysteines was observed to proceed through a relatively high-energy transition state (20.4 kcal mol$^{-1}$) and to be endergonic by 9.0 kcal mol$^{-1}$, in contrast to the 12–13 kcal mol$^{-1}$ barrier and 9–10 kcal mol$^{-1}$ exergonicity we computed for the related addition of a thiol to the MerB active site. Whereas the change in activation energy may be attributed to our neglect of the surrounding protein environment and to the absence, in the MerA study, of a direct proton transfer from the attacking thiol to the leaving Cys residue, further analysis points to another reason. Indeed, the QM-only results reported by Lian et al. in their Fig. 7 show that neglect of the electrostatic influence of the protein brings the activation energy down to 12.6 kcal mol$^{-1}$ (in perfect agreement with our data) but only lowers the reaction energy by 10 kcal mol$^{-1}$ (instead of the 19 kcal mol$^{-1}$ computed for the MerB reaction in our study). This observation allows us to attribute this 9 kcal mol$^{-1}$ energy difference to the additional interaction, in MerB, of Asp99 with the mercury ion. The influence of Asp99 is also noticeable in the steps involving thiolate addition to mercury, which occur without a barrier in MerB but have an activation energy of 9 kcal mol$^{-1}$ in the QM-only MerA model and (in a smaller extent) in the removal of a cysteine from a thiolate-attacked mercury, which has an activation energy of 11 kcal mol$^{-1}$ in the QM-only MerA model, compared to 15 kcal mol$^{-1}$ in MerB.

In spite of the similarity of the kinetic barriers of the different tested pathways, we were able to discriminate between the reaction pathways by comparing the kinetic profiles predicted by the quantum-chemical computations with the wealth of experimental data obtained by *Hong et al. (2010)* and *Benison et al. (2004)*. This analysis strongly suggests that *in vivo* the thiolate-only pathway is operative, and the Asp-assisted pathway (as well as the conversion of intermediates of the thiolate pathway into intermediates of the Cys-assisted pathway) is prevented by steric factors absent from our model and related to the precise geometry of the organomercurial binding-pocket. Considering the success we obtained with the combined used of quantum chemical and kinetic simulations, we strongly recommend that kinetic simulations be used (in addition to QM methods and experimental analysis of reaction time-coursed) whenever the analysis of reaction mechanism by quantum chemical methods does not afford a clearly preferred pathway.

## ACKNOWLEDGEMENTS

The authors thank Jan H. Jensen (University of Copenhagen) for helpful comments on this manuscript and James G. Omichinski (University of Montréal) for extensive discussions on the mechanism of MerB inhibition by DTT.

### Funding

Some portions of the work were performed using computational resources acquired under project PTDC/QUI-QUI/111288/2009, funded by the Portuguese Fundação para a Ciência e Tecnologia and FEDER through Programa Operacional Factores de Competitividade–COMPETE. The funders had no role in study design, data collection and analysis, decision to publish, or preparation of the manuscript.

### Grant Disclosures

The following grant information was disclosed by the authors:
Portuguese Fundação para a Ciência e Tecnologia: PTDC/QUI-QUI/111288/2009.
FEDER through Programa Operacional Factores de Competitividade–COMPETE.

### Competing Interests

The authors declare there are no competing interests.

### Author Contributions

- Pedro J. Silva conceived and designed the experiments, performed the experiments, analyzed the data, contributed reagents/materials/analysis tools, wrote the paper, prepared figures and/or tables, reviewed drafts of the paper.
- Viviana Rodrigues performed the experiments, analyzed the data.

### Data Deposition

The following information was supplied regarding the deposition of related data:

Complete data files for the quantum-chemical computations (including input, output and checkkpoint files) are available at:

http://dx.doi.org/10.6084/m9.figshare.991852.

The source code, compiled Windows executable and outputs of the kinetic simulation program used are available at:

http://dx.doi.org/10.6084/m9.figshare.1433993.

### Supplemental Information

Supplemental information for this article can be found online at http://dx.doi.org/10.7717/peerj.1127#supplemental-information.

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
