# Peer review of "Mechanistic pathways of mercury removal from the organomercurial lyase active site"

_PeerJ, doi:10.7717/peerj.1127_

## Round 0.1 · original submission · Major Revisions

· Academic Editor

Major Revisions

Please provide a point-to-point response letter to the reviewers' concerns.

Reviewer 1 ·

Basic reporting

In this paper, the author describe an computational MP2/CBS//B3PW91/6-31G(d) study of the final steps of the reaction of removing the methane from methylmercury catalyzed by MerB. This reaction steps occur in a complex potential energy surface where several distinct pathways are accessible and may operate concurrently. The topic is interesting, however, there are still questions need to be addressed.

Experimental design

Please specify what is MP2/CBS//B3PW91/6-31G(d) study?

Validity of the findings

What is the novelty of this study, compared with the recently computed mechanism of the related enzyme MerA further underline the important role of Asp99 in the energetics of the MerB reaction?

Additional comments

In this paper, the author describe an computational MP2/CBS//B3PW91/6-31G(d) study of the final steps of the reaction of removing the methane from methylmercury catalyzed by MerB. This reaction steps occur in a complex potential energy surface where several distinct pathways are accessible and may operate concurrently. The topic is interesting, however, there are still questions need to be addressed.
1. Please specify what is MP2/CBS//B3PW91/6-31G(d) study?
2.What is the novelty of this study, compared with the recently computed mechanism of the related enzyme MerA further underline the important role of Asp99 in the energetics of the MerB reaction?
3. Please be aware of the grammar in the context.

Reviewer 2 ·

Basic reporting

Introduction:
The purpose of this study implies to investigate the mechanistic pathway of mercury removal during the final steps of the reaction catalyzed by MerB as well as the regeneration of the active site for a new round of catalysis. Although the research gap based on the previous studies was clearly pointed out in the introduction, there is a lack of precise statement of purpose.

Methods and results:
Based on my understanding, all the methods and results were built on theoretical analysis via computer software. Lack of experimental data compensates the theoretical results.

Discussion:
The length of discussion section is un-proportionately shorter than results section. Discussion in depth is needed to address the important role of Asp99 in the energetics of MerB reaction. This manuscript seems to be uncompleted. No conclusion/summary or future perspectives directly or indirectly were included at the end which left the audiences a strong sense of an uncompleted paper.

References:
Lack of updated references. outdated references from 1986, 1997, etc. play a large part in this section.

Overall, lay language is excessively used in this manuscript (for example, lines 250, 259, 264). The whole manuscript could be written more precisely.

Experimental design

Please see above

Validity of the findings

Please see above

Additional comments

Please see above

---

## Round 0.2 · Minor Revisions

· Academic Editor

Minor Revisions

Please respond to the reviewer's comment. Thanks.

Reviewer 2 ·

Basic reporting

Overall, the revised manuscript was significantly improved. Authors did a job to address the reviewers' questions and concerns. Particularly, the added kinetic simulations strengthen the quality of results. However, there is a minor concern here. I could not find the following reference at the reference section. Line 221"Indeed, preliminary experimental evidence (J. omichinski, personal communication)"

Experimental design

"No Comments"

Validity of the findings

"No Comments"

Additional comments

Overall, the revised manuscript was significantly improved. Authors did a job to address the reviewers' questions and concerns. Particularly, the added kinetic simulations strengthen the quality of results. However, there is a minor concern here. I could not find the following reference at the reference section. Line 221"Indeed, preliminary experimental evidence (J. omichinski, personal communication)"

---

## Round 0.3 · accepted · Accept

· Academic Editor

Accept

Congratulations and thank you for your support of PeerJ.